

# SpeciesPrimer: a bioinformatics pipeline dedicated to the design of qPCR primers for the quantification of bacterial species

Matthias Dreier[1,2], Hélène Berthoud[1], Noam Shani[1], Daniel Wechsler[1] and Pilar Junier[2]

[1] Agroscope, Bern, Switzerland
[2] Laboratory of Microbiology, University of Neuchâtel, Neuchâtel, Switzerland

Corresponding author
Matthias Dreier,
matthias.dreier@agroscope.admin.ch,
matthias.dreier@unine.ch

## ABSTRACT

**Background**. Quantitative real-time PCR (qPCR) is a well-established method for detecting and quantifying bacteria, and it is progressively replacing culture-based diagnostic methods in food microbiology. High-throughput qPCR using microfluidics brings further advantages by providing faster results, decreasing the costs per sample and reducing errors due to automatic distribution of samples and reagents. In order to develop a high-throughput qPCR approach for the rapid and cost-efficient quantification of microbial species in complex systems such as fermented foods (for instance, cheese), the preliminary setup of qPCR assays working efficiently under identical PCR conditions is required. Identification of target-specific nucleotide sequences and design of specific primers are the most challenging steps in this process. To date, most available tools for primer design require either laborious manual manipulation or high-performance computing systems.

**Results**. We developed the SpeciesPrimer pipeline for automated high-throughput screening of species-specific target regions and the design of dedicated primers. Using SpeciesPrimer, specific primers were designed for four bacterial species of importance in cheese quality control, namely *Enterococcus faecium*, *Enterococcus faecalis*, *Pediococcus acidilactici* and *Pediococcus pentosaceus*. Selected primers were first evaluated *in silico* and subsequently *in vitro* using DNA from pure cultures of a variety of strains found in dairy products. Specific qPCR assays were developed and validated, satisfying the criteria of inclusivity, exclusivity and amplification efficiencies.

**Conclusion**. In this work, we present the SpeciesPrimer pipeline, a tool to design species-specific primers for the detection and quantification of bacterial species. We use SpeciesPrimer to design qPCR assays for four bacterial species and describe a workflow to evaluate the designed primers. SpeciesPrimer facilitates efficient primer design for species-specific quantification, paving the way for a fast and accurate quantitative investigation of microbial communities.

## INTRODUCTION

Quantitative real-time PCR (qPCR) is a well-established method for the detection and quantification of bacteria in microbiology, for instance in the context of pathogen detection in clinical and veterinary diagnostics and food safety (*Cremonesi et al., 2014*; *Curran et al., 2007*; *Garrido-Maestu et al., 2018*; *Ramirez et al., 2009*). Culture-based diagnostic methods are progressively being replaced by qPCR due to advantages such as faster results, more specific detection, and the ability to detect sub-dominant populations (*Postollec et al., 2011*). High-throughput microfluidic qPCR brings further advantages including the fast generation of results, a lower cost per sample and fewer errors due to automatic distribution of samples and reagents. However, in order to work efficiently, high-throughput qPCR systems use identical PCR chemistry and PCR conditions for all reactions taking place on a single chip. Therefore, existing qPCR assays are often not suitable and new primers have to be designed (*Hermann-Bank et al., 2013*; *Ishii, Segawa & Okabe, 2013*; *Kleyer, Tecon & Or, 2017*).

The main challenges for the successful development of any qPCR assay are the identification of a specific target nucleotide sequence and the design of primers that bind exclusively to that target sequence. Before microbial draft genomes became widely available, the 16S rRNA gene was frequently used as a target sequence. However, the regions that are targeted in the 16S rRNA gene often do not provide sufficient resolution to differentiate between closely related bacterial species (*Moyaert et al., 2008*; *Torriani, Felis & Dellaglio, 2001*; *Wang et al., 2007*). Further, housekeeping genes such as, for instance, *tuf*, *recA* and *pheS*, were successfully used as target sequences for a variety of bacterial species in fermented foods (*Falentin et al., 2010*; *Masco et al., 2007*; *Scheirlinck et al., 2009*). Today, the steadily increasing number of prokaryotic draft genomes facilitates the identification of new and unique target regions. This, in combination with the increased computing power, makes it now possible to screen and compare hundreds of genomes and to predict unique target sequences in a relatively short time.

Various commercial and open source programs facilitate the design of specific primers for a target sequence, such as the standard tools Primer3 and Primer-BLAST (*Untergasser et al., 2012*; *Ye et al., 2012*). Primer3 predicts suitable PCR primers for an input target sequence, while Primer-BLAST combines Primer3 with a BLAST search in a selected nucleotide sequence database to assess the specificity of the primers for the target sequence. Table 1 provides an overview of the features of different primer design tools and pipelines. PrimerMiner (*Elbrecht, Leese & Bunce, 2017*) is a tool that automatically downloads sequences of marker genes for taxonomic groups specified by the user and creates alignments and consensus sequences as target sequences for the design of degenerate primers. PrimerServer (*Zhu et al., 2017*) allows to design primers for multiple sites across a whole genome sequence and performs a specificity check. Tools and pipelines that encompass both the identification of target sequences from bacterial draft genomes and the design of primer candidates include, for instance, RUCS, the find_differential_primers (fdp) pipeline and TOPSI (*Pritchard et al., 2012*; *Thomsen et al., 2017*; *Vijaya Satya et al., 2010*). RUCS is able to identify unique core sequences in a positive set of genomes

(target) compared to a negative set of genomes (non-target). It designs primers for the core sequences and validates them with an *in silico* PCR validation method against the positive and negative reference sets. Similarly, the fdp pipeline designs primers for a set of positive genomes and, further, allows to extract primers specific to subclasses of the positive set and performs specificity check against a negative set of genomes. TOPSI is an automated high-throughput pipeline for the design of primers, primarily developed for pathogen-diagnostic assays. It identifies sequences present in all input genomes and designs specific primers accordingly.

We aimed to design a series of primers that function with the same qPCR cycling conditions and primer concentrations for later usage in a high-throughput microfluidic qPCR platform. RUCS, fdp and TOPSI can be used to design species-specific primers and offer high-throughput primer design. However, TOPSI could not be used because no Linux-based cluster was available. RUCS and fdp were initially not able to design primers for all our target species. Therefore, these pipelines were not suitable for our high-throughput approach.

This study presents a new pipeline named SpeciesPrimer developed for automated high-throughput screening for species-specific target regions combined with the design of primer candidates for these sequences. The process of primer design is fully automated from the download of bacterial genomes to the quality control of primer candidates. The pipeline runs on a standard computer with a multi-core processor and a minimum of 16 GB RAM. We have applied the SpeciesPrimer pipeline to a set of four bacterial species occurring in cheese and other dairy products and validated the primers *in silico* and *in vitro* by performing qPCR experiments with a variety of target and non-target strains.

## DESCRIPTION

### Overview

The SpeciesPrimer pipeline consists of three main parts (Table 2). First, genome assemblies are downloaded, annotated and then subjected to quality control. Second, a pan-genome analysis is performed to identify single copy core genes. Conserved sequences of these core genes are then extracted and the specificity for the target species is assessed. Finally, primers are designed for these species-specific conserved core gene sequences and subsequently evaluated in a primer quality control step. An overview of the features of the tools used for SpeciesPrimer can be found in Table S1.

### Part 1: Input genome assemblies

The minimal command line input for the pipeline is the species name. Further, a list of non-target species names can be specified (e.g., species found in the investigated ecosystem but that should not be detected in the specific qPCR assay). For downloading genome assemblies from the National Center for Biotechnology Information (NCBI) automatically, a valid e-mail address is required for accessing the NCBI E-utilities services (*Sayers, 2009*). The pipeline works with a pre-formatted NCBI BLAST database (nt), containing partially non-redundant nucleotide sequences. A local copy of the nt database is required. It can be downloaded from NCBI using the update_blastdb.pl script from

Dreier et al. (2020), *PeerJ*, DOI 10.7717/peerj.8544

**Table 1  Overview of the features of different primer design tools and pipelines.**

| Tool<br>Reference | RUCS<br>*Thomsen et al.*<br>*(2017)* | fdp<br>*Pritchard et al.*<br>*(2012)* | TOPSI<br>*Vijaya Satya et al.*<br>*(2010)* | Species-Primer<br>(this study) | Primer-Miner<br>*Elbrecht, Leese &*<br>*Bunce (2017)* | Primer-Server<br>*Zhu et al.*<br>*(2017)* | Primer-BLAST<br>*Ye et al.*<br>*(2012)* |
|---|---|---|---|---|---|---|---|
| Primer specificity | Bacterial strains / species | | | Bacterial species | Taxonomic groups | Input sequence | |
| **Inputs** | | | | | | | |
|   Taxonomic group(s) | – | – | – | Species | Order, Family | – | – |
|   Target gene(s) | – | – | – | – | x | – | x |
|   Genome assemblies | x | x | x | x | – | – | – |
|   Target sequences | – | – | – | – | x | x | x |
|   Primer sequences | x | x | – | – | x | x | x |
| **Automatic download of input sequences** | – | – | – | x | x | – | – |
| **Identification of target sequences** | x | x | x | x | – | – | – |
| **Identification of conserved regions** | x | – | x | x | x | – | – |
| **Primer design** | x | x | x | x | – | x | x |
| **Specificity check** | | | | | | | |
|   Target sequences | Input sequences | Input sequences | BLAST DB | BLAST DB | – | – | BLAST DB |
|   Primer | Input sequences | BLAST DB | BLAST DB | BLAST DB | Alignment | BLAST DB | BLAST DB |
| **Primer quality control** | x | x | x | x | – | x | |
|   Primer3 cutoffs | x | x | x | x | – | x | x |
|   Primer dimer | – | – | – | x | – | – | – |
|   Hairpin | – | – | – | – | – | – | – |
|   Amplicon secondary structures | – | – | – | x | – | – | – |

Dreier et al. (2020), *PeerJ*, DOI 10.7717/peerj.8544

**Table 1** (*continued*)

| Tool<br>Reference | RUCS<br>*Thomsen et al.<br>(2017)* | fdp<br>*Pritchard et al.<br>(2012)* | TOPSI<br>*Vijaya Satya et al.<br>(2010)* | Species-Primer<br>(this study) | Primer-Miner<br>*Elbrecht, Leese &<br>Bunce (2017)* | Primer-Server<br>*Zhu et al.<br>(2017)* | Primer-BLAST<br>*Ye et al.<br>(2012)* |
|---|---|---|---|---|---|---|---|
| High-throughput primer design | x | x | x | x | – | x | – |
| Batch processing | – | – | – | Full runs | Download | – | – |
| Works on standard computers | x | x | – | x | x | x | – |
| Graphic user interface | x | – | – | x | – | x | x |
| Web service | x | – | * | – | – | x | x |

**Notes.**

fdp, find_differential_primers; x, Feature supported; –, Feature not supported; *, Access has to be requested; QC, Quality control; CDS, Coding sequences.

**Table 2  Overview of the SpeciesPrimer pipeline workflow and the used software.**

| Pipeline workflow | Tools (Version[a]) | Reference |
|---|---|---|
| Input genome assemblies | | |
|    - download | NCBI Entrez (Biopython 1.73) | *Cock et al. (2009)*, *Sayers (2009)* |
|    - annotation | Prokka (1.13.7) | *Seemann (2014)* |
|    - quality control | BLAST+ (2.9.0+) | *Altschul et al. (1990)* |
| Core gene sequences | | |
|    - identification | Roary (3.12.0) | *Page et al. (2015)* |
|    - phylogeny | FastTree 2 (2.1.11) | *Price, Dehal & Arkin (2010)* |
|    - selection of conserved sequences | Prank (.150803) consambig (EMBOSS 6.6.0.0) GNU parallel (20161222) | *Löytynoja (2014)* *Rice, Longden & Bleasby (2000)* *Tange (2011)* |
|    - evaluation of specificity | BLAST+ | *Altschul et al. (1990)* |
| Primer | | |
|    - design | Primer3 (2.4.0) | *Untergasser et al. (2012)* |
|    - quality control | BLAST+, MFEprimer (2.0), MPprimer (1.5), Mfold (3.6) | *Altschul et al. (1990)* *Qu et al. (2012)* *Shen et al. (2010)* *Zuker, Mathews & Turner (1999)* |

**Notes.**

[a] Docker image June 13, 2019.

the BLAST+ package (*Altschul et al., 1990*), via FTP from the NCBI FTP server or with the pipeline script (getblastdb.py). The nt database, which consists of sequences from GenBank, EMBL (European Molecular Biology Laboratory) and DDBJ (DNA Data Bank of Japan), was selected because it has a large coverage of diverse sequences, but it is not as large as for example the refseq_genomic database (*Tao et al., 2011*). The evaluation of the specificity of the target sequence for the target species does not rely on small differences in the nucleotide sequence, but on the overall similarity. Therefore, even with one genome sequence per non-target species we would expect to find similarities in the core genes of the non-target species. Each additional genome of this species in a database would then allow finding more potential sequence similarities in shell genes, cloud genes and strain-specific genes. On the one hand, a more extensive database could better predict the specificity of a sequence, but on the other, it would increase the size of the database and the time required for the BLAST search.

The user-provided species name is used to search for genome assemblies in the NCBI database. The Biopython Entrez module (*Cock et al., 2009*) searches the NCBI taxonomic identity (taxid) for the target species in the taxonomy database and downloads the genome assembly summary report. Afterwards, SpeciesPrimer downloads the genome assemblies in FASTA format from the NCBI RefSeq FTP server using the links specified in the summary report. Finally, the downloaded genome assemblies are annotated with Prokka (*Seemann, 2014*).

The quality of the genome assemblies is a crucial factor for the pan-genome analysis. Genome assemblies deposited with the wrong taxonomic label or low-quality assemblies drastically reduce the number of identified core genes and of conserved sequences for

primer design. The initial quality control step is intended to remove such assemblies from the subsequent analysis. For the verification of the taxonomic classification, the user can choose one or several genes from five conserved housekeeping genes (16S rRNA, *tuf, recA, dnaK* and *pheS*). Genome assemblies without an annotation for the specified conserved housekeeping genes or genome assemblies consisting of more than 500 contigs are removed from the downstream pan-genome analysis. The sequences of the specified conserved housekeeping genes are blasted against the local nt database. Genome assemblies pass the quality control if the best BLAST hit for all sequences is a sequence arising from the target species.

## Part 2: Identification of target sequences for primer design

A pan-genome analysis is performed using Roary (*Page et al., 2015*) to identify the core genes of the target species. Based on the results of the pan-genome analysis, single copy core genes are identified. The gene_presence_absence.csv produced by Roary reports the presence (or absence) of every annotated gene for every input genome assembly. Single copy core genes are the genes for which the number of assemblies harboring the sequence and the number of total identified sequences equals the number of total input assemblies. An sqlite3 database containing all annotated sequences of all assemblies is compiled using the DBGenerator.py script from the Microbial Genomics Lab GitHub repository (https://github.com/microgenomics/tutorials). This database is queried for single copy core genes and the nucleotide sequences are saved in multi-FASTA format. Each multi-FASTA file contains the sequences of one single copy core gene from each input genome assembly. These sequences are aligned using the probabilistic multiple alignment program Prank (*Löytynoja, 2014*). A consensus sequence with ambiguous bases is then created using the consambig function from the EMBOSS package (*Rice, Longden & Bleasby, 2000*). The alignments and extraction of the consensus sequences are performed in parallel for several core genes using GNU parallel (*Tange, 2011*). Continuous consensus sequences longer than the minimal PCR product length, harboring less than two ambiguous bases in the range of 20 bases are used for the subsequent steps of the pipeline.

These conserved consensus sequences are used for a BLAST search against the local nt database using the discontiguous BLAST algorithm and an *e*-value cutoff of 500. For all hits in the BLAST results, the species name is extracted from the sequence description and compared with the names in the species list (non-target species). If any species name in the species list matches a hit in the BLAST results the corresponding query sequence is discarded, otherwise the sequence is classified as specific for the target and considered for primer design.

## Part 3: Primer design

Primer3 is used to design primers for the unique single copy core gene sequences. As pipeline default, the optimum primer melting temperature is set to 60 °C and the maximal primer length is set to 26 bases. All other settings are the default settings of the primer3web version (http://primer3.ut.ee, accessed November 29, 2018). The minimal and maximal amplicon size of the PCR product can be specified individually for every target species

through the command line options. The other parameters for Primer3 cannot be changed individually, but the general Primer3 settings can be changed by modifying the Primer3 settings file.

The primer quality control consists of three parts, an *in silico* PCR to evaluate the specificity of the primer for the template, an estimation of secondary structures of the amplicon sequence and the estimation of the potential to form primer dimers. The specificity check (Fig. 1) for each primer pair is performed with MFEprimer-2.0 (*Qu et al., 2012*). For the evaluation of the specificity, three indexed databases are generated: the target template database, the non-target sequence database and the target genome database. The target template database consists of the unique conserved core gene sequences used as template for primer design. The non-target sequence database is compiled from sequences of non-target species that show similarities to the primer sequences. To identify these sequences, a BLAST search with all primers against the local nt database is performed. BLAST hits with a species name in the description matching a name in the user-specified non-target species list are selected. These selected sequences and 4000 base pairs up- and downstream are extracted from the nt database using the blastdbcmd tool. The target genome database is composed of maximal 10 of the input genome assemblies. If the assembly summary report from the automatic download of genome assemblies from NCBI is available, the genome assemblies as complete as possible are preferred (assembly status: complete >chromosome >scaffold >contig). The target sequence database is used to evaluate the maximum primer pair coverage (PPC, maximum value = 100), a value used by MFEprimer-2.0 to score the ability of the primer pair to bind to a DNA template. All primer pairs with a PPC value lower than the specified threshold (mfethreshold, default = 90) for their template are excluded. Next, MFEprimer-2.0 is used to score the binding of the primer pairs to the sequences of the non-target sequence and the target genome database. The difference of the PPC for the DNA template and the specified threshold ($\Delta$threshold = PPC −mfethreshold) is used as a threshold for the maximum PPC value a primer pair is allowed to have for a non-target sequence. Strong secondary structures at the 5′- or the 3′- end of the PCR product could impair efficient primer binding. Therefore, the PCR products of the primer pairs are submitted to mfold (*Zuker, Mathews & Turner, 1999*) to exclude PCR products with strong secondary structures at the annealing temperature of 60 °C. Moreover, as primer dimers can yield unspecific signals during the qPCR run, the 3′- ends of the primer pairs are checked for their potential to form homo- or hetero-dimers using a Perl script (MPprimer_dimer_check.pl) from MPprimer (*Shen et al., 2010*).

The pipeline output is a list containing the primer name, primer pair coverage (MFEprimer) and penalty values, primer and template sequences and melting temperatures (Primer3). Further, a report of the genome assembly quality control, a file containing the pipeline run statistics, the core gene alignment and the phylogeny in newick format can be found in the output directory.

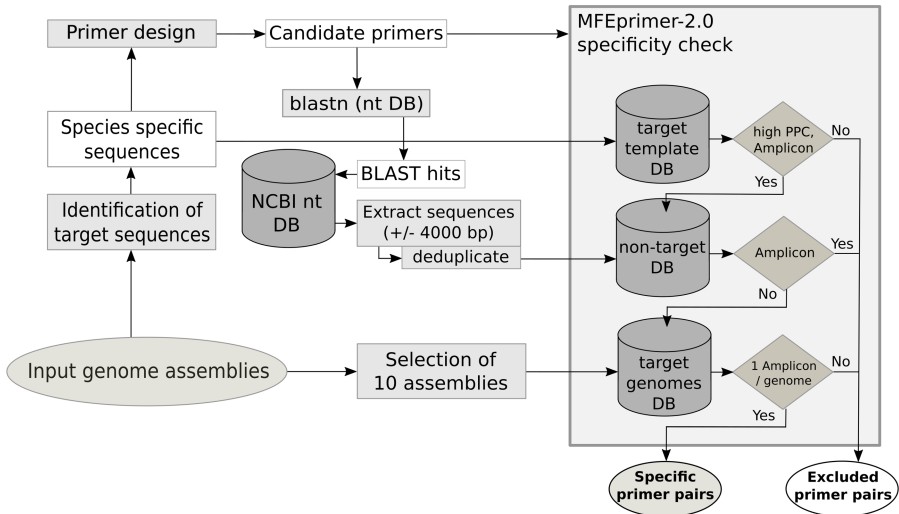

**Figure 1** Schematic workflow of the database creation and the specificity check using MFEprimer-2.0.

## MATERIALS & METHODS

### Primer design

SpeciesPrimer pipeline runs were performed on a virtual machine (Oracle VM VirtualBox 5.2.8) with Ubuntu 16.04 (64-bit) and Docker installed, using 22 of 24 logical processors from two Intel Xeon E5-2643 CPUs, 32 GB of RAM, a solid-state drive and a LAN Internet connection. To show the performance of the SpeciesPrimer pipeline on consumer hardware, the runs were repeated on a laptop with an Ubuntu 16.04 (64-bit) operating system, an i7-3610QM CPU (8 logical processors), 8 GB RAM, a solid-state drive and a wireless LAN Internet connection. The used Docker image is available at https://hub.docker.com/r/biologger/speciesprimer.

The species list consisted of 259 species and subspecies names detected in dairy products, namely from species names collected from data of 16S rRNA amplicon sequencing studies in milk and cheese varieties (Marco Meola Agroscope, pers. comm.) and dairy-related bacteria from the list of bacterial species and subspecies with technological beneficial use in food products (*Almeida et al., 2014*).

The SpeciesPrimer pipeline was run with the input genome assemblies, parameters and the species list specified in the supplemental information (Dataset S1). Genome assemblies from the Agroscope Culture Collection were included for the Pediococci.

### *In silico* validation

For the *in silico* validation, PCR products for the designed primer pairs were used for an online BLAST search against the RefSeq Genomes Database (refseq_genomes) limited to bacterial genomes. The search was performed by qblast (Biopython), using blastn, the maximum hitlist size was set to 5000 and the expect threshold (*e*-value) was set to 500.

Primer pairs were tested for specificity using online Primer-BLAST. The primers were blasted against the nucleotide collection BLAST database (nr) limited to sequences from
bacteria. The nr (non-redundant nucleotide) database was chosen to get the broadest coverage for the BLAST search. Default settings were used, except for the primer specificity stringency that was set to ignore targets that have nine or more mismatches to the primer.

### *In vitro* validation

The inclusivity of the primer pairs was assayed by performing qPCR with 2 ng DNA of 21 to 25 strains of the target species in technical duplicates. The PCR efficiency was examined by ten-fold dilution series of the type strain DNA in a range from $10^6$ to $10^1$ genome copies per reaction. DNA concentration for the corresponding number of genome copies was estimated by taking the genome size of the type strain (https://www.ncbi.nlm.nih.gov/genome) and an average weight of $1.096 \cdot 10^{-21}$ g per base pair.

The exclusivity of the primer pairs was assayed by performing qPCR in technical duplicates with 2 ng DNA from various bacterial species found in dairy products. Because the number of samples per run was limited, four separate runs were required to measure all non-target strains. In each run three strains of the target species (positive control) and a no template control were included.

### Bacterial strains

Strains stored within the Agroscope Culture Collection at $-80$ °C in sterile reconstituted skim milk powder (10%, w/v), were reactivated and cultivated according to the conditions specified in Dataset S2.

### DNA extraction

Unless otherwise noted, all reagents were purchased from Merck, Darmstadt, Germany.

Bacterial pellets harvested from 1 ml culture by centrifugation (10,000× g, 5 min, room temperature) were used for DNA extraction. For a pre-lysis treatment, the bacterial cells were incubated in 1 ml of 50 mM sodium hydroxide for 15 min at room temperature. Afterwards cells were collected by centrifugation (10,000× g, 5 min, room temperature) and then treated with lysozyme (2.5 mg/ml dissolved in 100 mM Tris(hydroxymethyl)aminomethane, 10 mM ethylendiaminetetraacetic acid (EDTA; Calbiochem, San Diego, USA), 25% (w/v) sucrose, pH 8.0) for 1 h at 37 °C. After the pre-lysis treatment, the bacterial cells were collected by centrifugation (10,000× g, 5 min, room temperature). Cell lysis and genomic DNA extraction was performed using the EZ1 DNA Tissue kit and a BioRobot® EZ1 workstation (Qiagen, Hilden, Germany) according to the manufacturer's instructions and eluted in a volume of 100 µl. The DNA concentration was measured using a NanoDrop® ND-1000 Spectrophotometer (NanoDrop Technologies, Thermo Fisher Scientific, Waltham, MA, USA).

### Quantitative real-time PCR

The qPCR assays were performed in a total reaction mix volume of 12 µl containing 6 µl 2x SsoFast™ EvaGreen® Supermix with low ROX (Biorad, Cressier, Switzerland), 500 nM of forward and reverse primers, respectively, and 2 µl of DNA. Each sample was measured in technical duplicates. The qPCR cycling conditions were an initial denaturation at 95 °C

for 1 min followed by 35 cycles of 95 °C for 5 s and 60 °C for 1 min. For the melting curve analysis, a gradient from 60–95 °C with 1 °C steps per 3 s was performed. All qPCR assays were run on a Corbett Rotor-Gene 3000 (Qiagen). The analysis was performed using Rotor-Gene 6000 Software 1.7 with dynamic tube normalization and a threshold of 0.05 for quantification cycle (Cq) value calculation, the five first cycles were ignored for the determination of the Cq values. The peak calling threshold for the melt curve analysis was set to -2 dF/dT and a temperature threshold was set 2 °C lower than the positive control peak.

## Phylogeny and average nucleotide identity calculations

The phylogeny was created with Roary and FastTree 2 during the pipeline runs and iTOL (*Letunic & Bork, 2019*) was used to visualize the tree. Average nucleotide identity (ANI) calculations were performed with pyani v0.2.9 (*Pritchard et al., 2016*) using the ANIm method. The heatmap was created from the ANIm_percentage_identity.tab output file using the clustermap function of the python seaborn module and modified color bar settings from pyani. For the color bars on top and on the left of the heatmap, the assemblies were assigned to the same color as in the phylogeny tree. Row and column names (genome assembly accessions) can be found in Dataset S3.

## Comparison of primer design pipelines

The positive genome sets for RUCS and fdp were the same genome assemblies used for the SpeciesPrimer pipeline. SpeciesPrimer uses by default the NCBI nt database and the species list for the specificity checks, whereas RUCS and fdp require a negative set of genomes. Therefore, a set of (representative) genome assemblies from NCBI was downloaded for the species from the species list. From these assemblies a BLAST database was prepared for SpeciesPrimer. The same genome assemblies, excluding the assembly of the target species, were used as a negative set for RUCS and to make a BLAST database for fdp. For both tools, the minimal and maximal PCR product size was set to 70 and 200, respectively. The tab separated config file for fdp was created using the assembly accession as name, the species as class and providing the absolute path of the genome assembly files. The script was started with the blastdb option to provide the path to the previously prepared BLAST database with the non-target genome sequences. For RUCS the entry point full was selected and the annotation of the target sequences was omitted. SpeciesPrimer was configured to run with the custom BLAST database, without a species list and the download and annotation step for the genome assemblies was omitted to provide comparable running conditions. The accessions of the input genome assemblies and the commands used can be found in Dataset S4. Primers used for a specificity check using Primer-BLAST (nr database limited to sequences from bacteria) were the two primer pairs with the best score in the results_best.tsv files (RUCS), the two best ranked primer pairs for SpeciesPrimer and the primers reported in the universal_primers.eprimer3 files (fdp).

## RESULTS

### Primer design

The SpeciesPrimer pipeline runs were completed in two to eight hours, excluding the time required for downloading and annotation of the genome assemblies. Depending on the number of genome assemblies, downloading and annotation of the genome assemblies took from 24 min (27) to 12 h 27 min (575). The average time for downloading and annotation of single assemblies was two seconds and one minute six seconds, respectively. On the consumer laptop using a wireless LAN Internet connection the time required for the downloads has doubled, while the annotation took 1.8 times longer. The pipeline runs lasted in total three times longer and were completed in seven to 29 h. The analysis of the *Enterococcus faecalis*, *Enterococcus faecium*, *Pediococcus acidilactici* and *Pediococcus pentosaceus* assemblies resulted in 15, 2, 2 and 160 identified primer pair candidates, respectively (Table 3). The primer pair candidates for *E. faecalis* and *P. pentosaceus* were filtered for the highest primer pair coverage score (*E. faecalis*: 2; *P. pentosaceus*: 29); for *P. pentosaceus*, only the two primer pairs with the lowest primer pair penalty values were selected.

The phylogeny tree from the concatenated core genes of *E. faecium* shows the phylogenetic distance of two distinct groups of sequences, a main cluster with 531 sequences and a subcluster with 44 sequences (Fig. 2). The tree made with the concatenated core gene sequences of *P. acidilactici* shows the phylogenetic distance of one sequence from all other sequences (Fig. 3). From this observation, the existence of different taxonomic units was suspected. Calculation of the average nucleotide identity (ANI) has been proposed as a valuable tool to determine species boundaries (*Richter & Rossello-Mora, 2009*). Therefore, we performed ANI calculations for the genome assemblies and displayed the results in a clustered heatmap (Fig. 4). All genome assemblies show an alignment coverage of at least 60% to each other (Dataset S3), indicating they are correctly assigned at the genus level. The clustering of the *E. faecium* genome assemblies in Fig. 4 A shows two distinct clusters corresponding to the clusters in the phylogenetic tree (Fig. 2). The assemblies of the two clusters have ANIm values at the border of the species threshold cutoff as depicted by the white to light purple colored cells. Clustering of the *P. acidilactici* genome assemblies in Fig. 4 B shows three distinct clusters corresponding to the clusters in the phylogenetic tree (Fig. 3). The purple cells indicate that the assemblies of two larger clusters belong to the same species, while the assembly with the orange color bar has ANIm percentage identity values below the proposed species threshold cutoffs (95–96%) (*Kim et al., 2014*; *Richter & Rossello-Mora, 2009*) as indicated by the blue cells. *P. acidilactici* strain FAM 18987 should therefore probably be assigned to a new species or subspecies. However, for certain species lower boundary cutoffs might be reasonable (*Ciufo et al., 2018*). According to the current taxonomic classification, we proceeded with the assumption that these genome assemblies reflected the actual diversity of strains and thus included the assemblies for the primer design.

Two test cases were generated to exemplify the influence of the input genome assemblies on the pipeline results. Firstly, a single genome assembly with a wrong taxonomic label

**Table 3  Pipeline input and run statistics.** Two different computers were used to run the SpeciesPrimer pipeline depicted as high end desktop and consumer laptop. The high end desktop is running Ubuntu 16.04 in a virtual machine with two Xeon E5-2643 CPU's (22 logical processors), 32 GB RAM and a solid-state drive. The download of the genome assemblies was performed using a LAN connection. The consumer laptop is running on Ubuntu 16.04 with an i7-3610QM CPU (8 logical processors), 8 GB RAM and a solid-state drive. The download of the genome assemblies was performed using a wireless LAN connection.

| Species | E. faecalis | E. faecium | P. acidilactici | P. pentosaceus |
|---|---|---|---|---|
| **Pipeline input** | | | | |
| NCBI genomes | 390 | 575 | 9 | 14 |
| ACC genomes | 0 | 0 | 118 | 13 |
| **Total genome assemblies** | **390** | **575** | **127** | **27** |
| **Download and annotation (h:min)** | | | | |
| High end desktop | 9:04 | 12:27 | 1:55 | 0:24 |
| Consumer laptop | 15:56 | 22:18 | 3:10 | 0:42 |
| **Pipeline statistics** | | | | |
| Running time (h:min) | | | | |
| High end desktop | 6:11 | 8:05 | 1:55 | 4:25 |
| Consumer laptop | 19:52 | 28:56 | 6:59 | 6:47 |
| Core genes | 1375 | 1131 | 921 | 1341 |
| Single copy core genes | 632 | 563 | 641 | 889 |
| Conserved sequences | 1128 | 624 | 566 | 2782 |
| Species-specific sequences | 329 | 36 | 54 | 672 |
| Potential primer pairs | 89 | 4 | 7 | 632 |
| **Primer pairs after QC** | **15** | **2** | **2** | **160** |

**Notes.**

QC, primer quality control; ACC, Agroscope Culture Collection.

was used as input in addition to the correctly labelled genome assemblies. Introducing a genome assembly with a wrong taxonomic label (GCF_000415325.2, *E. faecalis*) into the pool of *E. faecium* genome assemblies resulted in a decrease of identified core genes (from 1131 to 43) and provided no species-specific sequence. Secondly, the genome assembly of the *P. acidilactici* strain (FAM 18987) that was distinct from the other assemblies in the phylogenetic tree and had ANI values below the species threshold cutoff was excluded from the pipeline run. This resulted in an increased number of identified core genes (from 921 to 1238), of species-specific sequences (from 54 to 516) and of reported primer pairs (from 2 to 53).

### *In silico* validation

Two parameters were selected as criteria for the primer validation using web-based BLAST. First, the BLAST hits for the predicted PCR product sequence should only match the target species. If sequences of other bacterial species matched to parts of the sequence, the corresponding primer pairs were discarded, unless more than three mismatches were found in each primer-binding region for the forward and reverse primers. Second, the primer binding sites in the target sequences were not allowed to have mismatches in the 3′-end region. The criterion for the primer validation by Primer-BLAST was that no predicted PCR products for other bacterial species had been reported by Primer-BLAST.

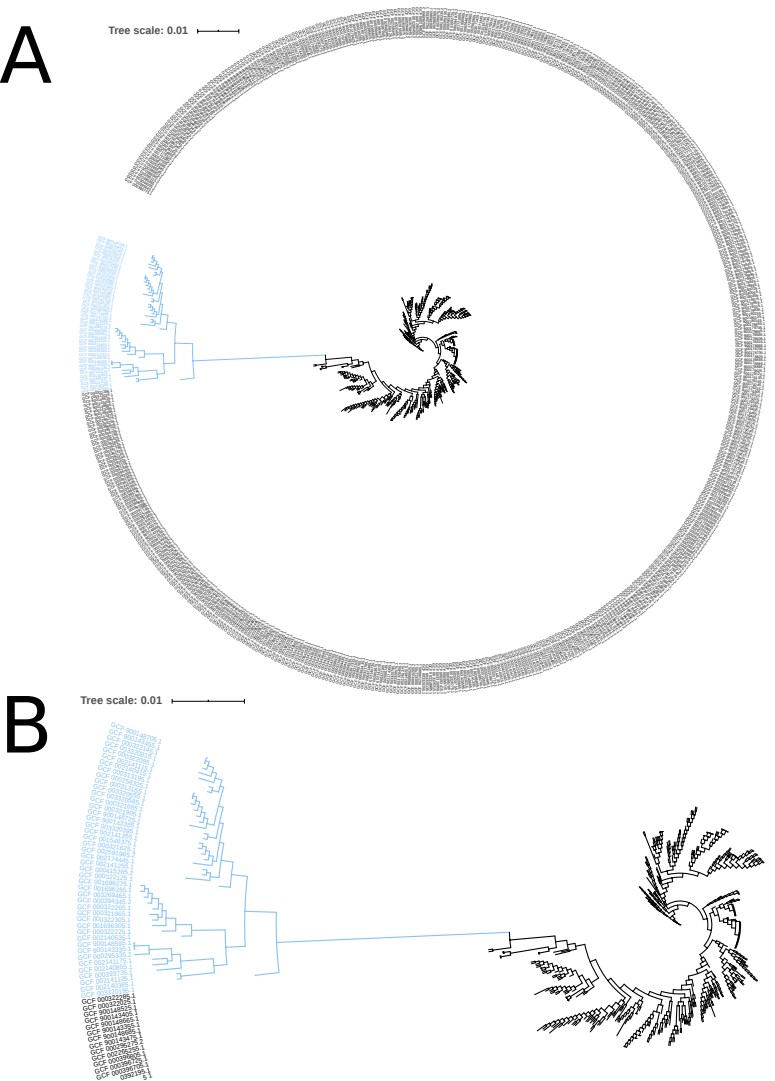

**Figure 2** **Phylogenetic tree based on the alignment of concatenated core genes of 575** *Enterococcus faecium* **genome assemblies.** (A) The main cluster with 531 sequences is depicted in black and the subcluster of 44 sequences in blue. (B) Enlarged view of the tree structure and the subcluster.

Primer pairs exclusively binding to the target sequence of the target species were classified as specific. The results of the *in silico* validation are summarized in Table 4. With the exception of Ec_faeca_g3060_1_P0 and Ec_faeci_cysS_3_P1, all primer pairs showed a perfect match to their target sequences. For primer pair Ec_faeca_g3060_1_P0, the first three nucleotides of one sequence out of 690 are missing in the forward primer-binding region. For Ec_faeci_cysS_3_P1, only one sequence out of 1058 aligned sequences showed a single nucleotide transition in the reverse primer-binding region (Dataset S5, page 2–3).

### *In vitro* validation

The specificity of the qPCR assays was assessed with 21 to 25 strains of the target species (inclusivity) and 120 non-target bacterial strains found in dairy products (exclusivity). The

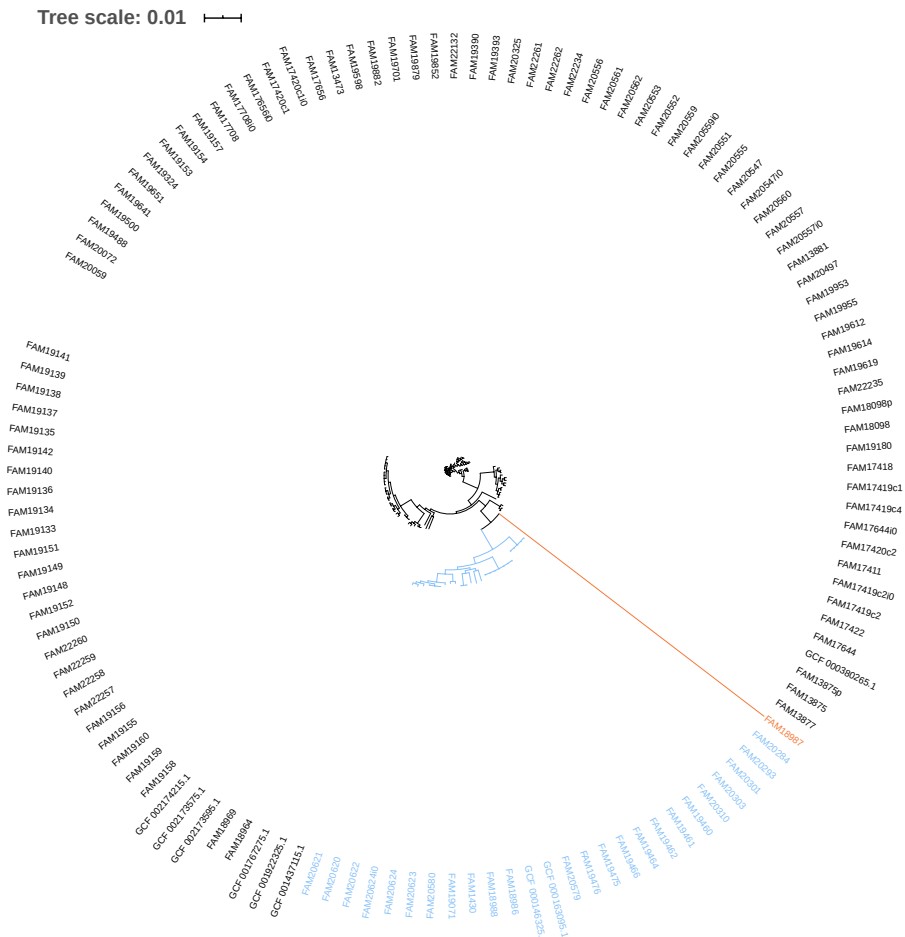

**Figure 3** **Phylogenetic tree based on the alignment of concatenated core genes of 127 *Pediococcus acidilactici* genome assemblies.** The main cluster with 100 sequences is depicted in black, a subcluster of 26 sequences in blue and the sequence with the largest phylogenetic distance in orange.

qPCR assay performance was assessed by 10-fold dilution series of type strain DNA from $10^6$ to $10^1$ copies per reaction. The results of the qPCR runs were interpreted as positive if both qPCR reactions (duplicates) reached the fluorescent threshold before quantification cycle 35 and the peak of the melting curve analysis was above the peak calling threshold (-2 dF/dT). A summary of the results is shown in Table 5. The primer sequences can be found in Table S2. The inclusivity of the qPCR assays was 100% for the assays Ec_faeca_acuI, Ec_faeca_g3060, Ec_faeci_cysS, Pd_acidi_asnS, Pd_acidi_g1164, Pd_pento_nagK and Pd_pento_g4364. Only one qPCR assay, Ec_faeci_purD was negative for one of the tested target strains.

Out of the 120 non-target strains analyzed to determine the exclusivity of the qPCR assays (Fig. 5), all strains were negative for Ec_faeca_acuI and Pd_acidi_asnS. The assay Pd_pento_nagK targeting *P. pentosaceus* was positive for two out of three tested *Leuconostoc lactis* strains, the fluorescence signal reached the threshold after Cq 26, and the melting curve analysis showed a peak at 85 °C, while the positive control samples for this assay

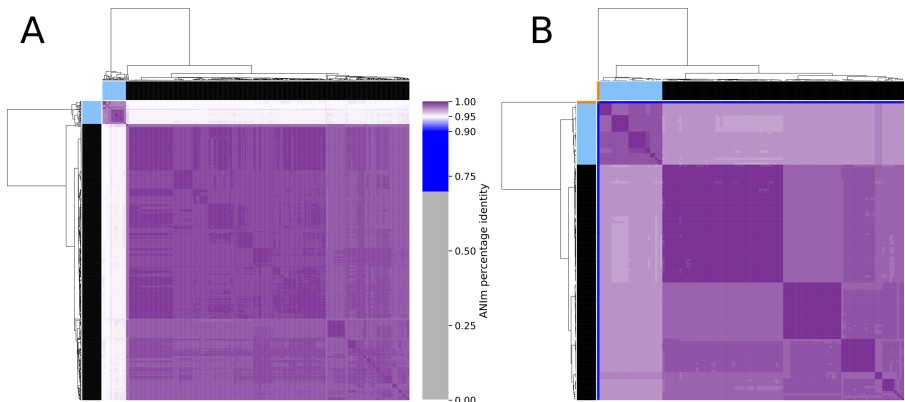

**Figure 4** **Clustered heatmap of ANIm percentage identity for (A) 575** *Enterococcus faecium* **and (B) 129** *Pediococcus acidilactici* **genome assemblies.** Purple colored cells in the heatmap correspond to ANIm percentage identity above 95%, color intensity fades towards the proposed species threshold cutoff. Blue colored cells are below this threshold indicating that the corresponding genome assembly does not belong to the same species. Color bars on top and on the left of the heatmap correspond to the clusters and the colors indicated in the phylogenetic trees (Figs. 2 and 3). Dendograms are based on single-linkage hierarchical clustering of the ANIm percentage identities. Row and column names can be found in Dataset S3.

**Table 4** **Summary of the in** *silico* **validation of the selected primer pairs.** Primer pair coverage (PPC) is a value used by MFEprimer-2.0 to score the ability of the primer pair to bind to a DNA template. The number of perfect matches of the primers to the primer binding region and the total number of target sequences are indicated in brackets.

| Target species | Primer pair | PPC | BLAST (perfect/total) | Primer-BLAST (perfect/total) |
|---|---|---|---|---|
| E. faecalis | Ec_faeca_acuI_1_P0 | 100 | specific (694/694) | specific (55/55) |
| | Ec_faeca_g3060_1_P0 | 100 | specific (689/690) | specific (55/55) |
| E. faecium | Ec_faeci_cysS_3_P1 | 96.7 | specific (1057/1058) | specific (148/148) |
| | Ec_faeci_purD_2_P0 | 93.3 | specific(1083/1083) | specific (148/148) |
| P. acidilactici | Pd_acidi_asnS_2_P0 | 90.1 | specific (19/19) | specific (11/11) |
| | Pd_acidi_g1164_1_P0 | 93.3 | specific (23/23) | specific (11/11) |
| P. pentosaceus | Pd_pento_nagK_1_P0 | 100 | specific (15/15) | specific (9/9) |
| | Pd_pento_g4364_1_P0 | 100 | specific (15/15) | specific (9/9) |

displayed a peak at 83.5 °C. Nine out of the 120 non-target strains were positive for the Ec_faeca_g3060 qPCR assay. For these samples the fluorescence signals reached the threshold after Cq 26 and had a melting curve peak at a higher temperature than the target PCR product. The assays Pd_acidi_g1164 and Pd_pento_g4364 were positive for five and eight non-target strains, respectively. Notably, all three tested *Lactobacillus paracasei* strains were positive for the Pd_acidi_g1164 assay, the fluorescence signal reached the threshold around Cq 21 and 22 and they showed a distinct melting curve peak at 86 °C.

The calculated efficiency of the qPCR assays was between 92 and 100%. The linear regression equations $(Cq = slope * \log(copies) + intercept)$ had slopes between -3.329 and

**Table 5** **Summarized results of the *in vitro* validation of the selected qPCR assays.** Inclusivity: Number of positive DNA samples / total number of target species DNA samples. Exclusivity: Number of DNA samples showing a fluorescence signal below quantification cycle 35 and a melting curve peak above the threshold / total number of non-target DNA samples. Calculated efficiency, slope, intercept and correlation coefficient ($R^2$) of the linear regression equation.

| Species | *E. faecalis* | | *E. faecium* | | *P. acidilactici* | | *P. pentosaceus* | |
|---|---|---|---|---|---|---|---|---|
| Target gene | *acuI* | g3060 | *cysS* | *purD* | *asnS* | g1164 | *nagK* | g4364 |
| Inclusivity | 22/22 | 22/22 | 25/25 | 24/25 | 21/21 | 21/21 | 25/25 | 25/25 |
| Exclusivity | 0/120 | 9/120 | 0/120 | 0/120 | 0/120 | 5/120 | 2/120 | 8/120 |
| Efficiency | 98% | 97% | 92% | 97% | 99% | 100% | 94% | 92% |
| Slope | −3.382 | −3.387 | −3.539 | −3.396 | −3.356 | −3.329 | −3.470 | −3.523 |
| Intercept | 32.107 | 32.694 | 32.006 | 31.051 | 30.835 | 30.282 | 32.286 | 33.211 |
| $R^2$ | 0.998 | 0.997 | 0.990 | 0.996 | 0.997 | 0.995 | 0.996 | 0.997 |

-3.523 and correlation coefficients of 0.990 or above. Dataset S6 contains the qPCR raw data and Dataset S7 a summary of the qPCR data.

## Comparison of primer design pipelines

The running times and the number of reported primer pairs of RUCS, the fdp pipeline, and SpeciesPrimer were compared. The download and annotation times were not considered since RUCS and fdp do not include this feature. RUCS and fdp were both able to design primer pairs for all four bacterial targets. The runs with RUCS were completed in two hours and 11 min to five hours and 20 min and between 107 and 629 primer candidates were reported. The specificity check using online Primer-BLAST showed that the best-ranked primer pair for each of the targets was specific and perfectly matched to the primer binding region for all targets in the nr database. The fdp runs were completed in two min to 17 h 44 min. Three primer pairs were reported for *E. faecium* and *P. acidilactici* and six primer pairs were reported for *E. faecalis* and *P. pentosaceus*. Primer-BLAST results indicate that the best primer pairs for all target species are specific. The best primer pair for *P. acidilactici* showed a two-nucleotide mismatch in the primer binding region of one target sequence. A one-nucleotide mismatch in the primer binding region of one target sequence was also observed in the primer pair for *E. faecalis* (Dataset S4, Primer-BLAST summary). The results of the SpeciesPrimer runs differ from the runs with the nt BLAST database presented in detail above. For the Enterococci the best reported primer pairs remain the same, while different primer pairs were ranked best for the Pediococci.

In summary, all pipelines were able to design species-specific primers for all of the four target species using the given input sequences. The results of the comparison are summarized in Table 6.

## DISCUSSION

After setup of the SpeciesPrimer Docker container, the download of the local BLAST database and the selection of the SpeciesPrimer run settings, no further manual handling was required to get primer pair candidates for all four bacterial species after a total time of 44 h and 30 min (high end desktop). The number of input genomes and subsequently the

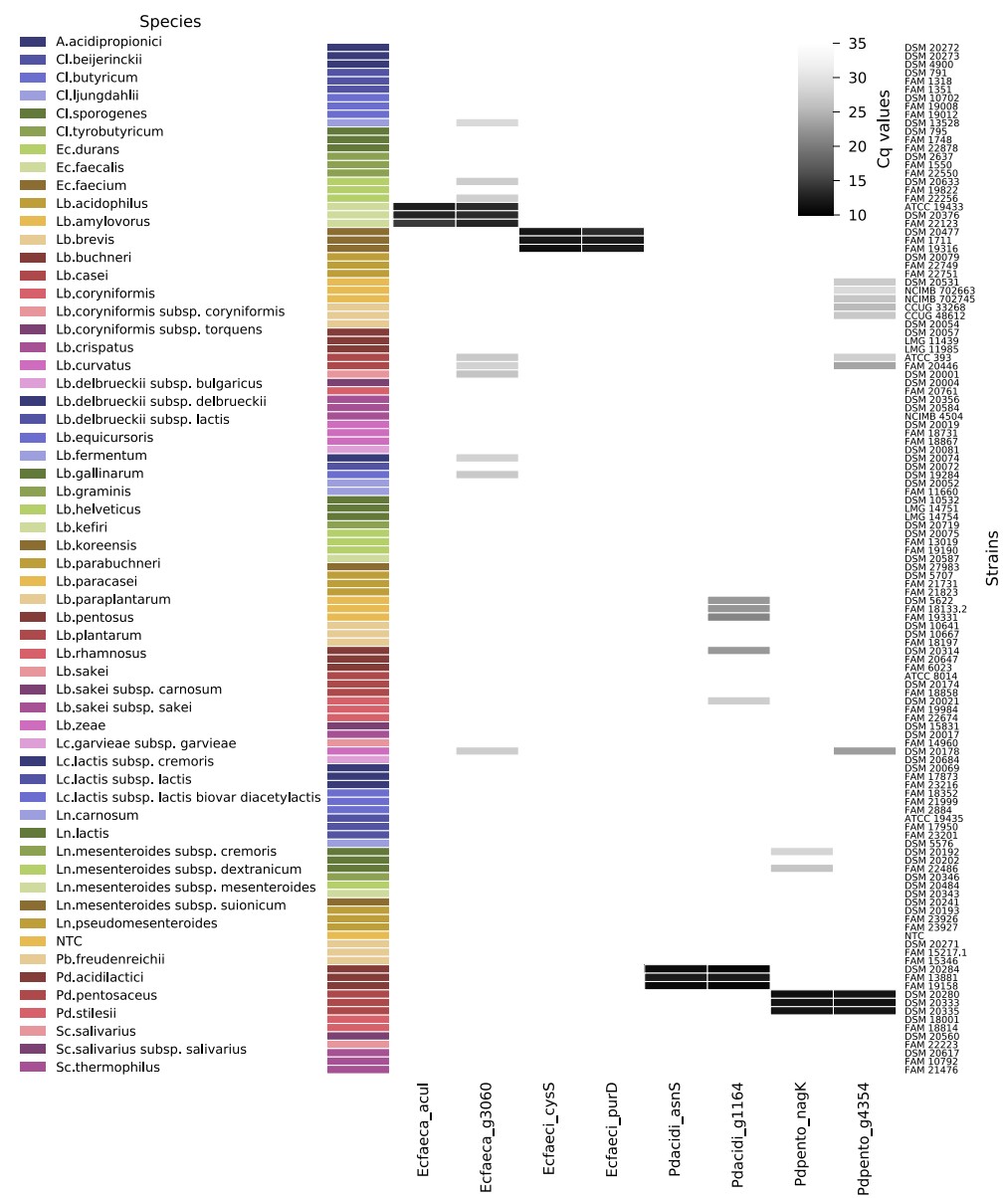

**Figure 5 qPCR assay quantification cycle heatmap.** Depicted are all tested non-target strains and their average quantification cycle (technical duplicates) if the melt curve peak was above the threshold. The gray shades represent the Cq values from 10 to 35 (if no fluorescent signal was measured the value was set to Cq 35). Abbreviations: A., *Acidipropionibacterium*; Cl., *Clostridium*; Lb., *Lactobacillus*; Ln., *Leuconostoc*; Pb, *Propionibacterium*; Pd., *Pediococcus*; Sc., *Streptococcus*; NTC, no template control.

number of retrieved primer pairs for the specificity check have the highest impact on speed. During the specificity check, blasting the primer sequences optimized for short sequences (blastn-short) and the subsequent compilation and indexing of the non-target sequence database are the most time consuming steps.

**Table 6  Comparison of different primer design pipelines.** The hardware used to run the pipelines was a high end desktop running Ubuntu 16.04 in a virtual machine with two Xeon E5-2643 CPU's (22 logical processors), 32 GB RAM and a solid-state drive. For the Primer-BLAST results, the number of perfect matches of the primers to the primer binding region and the total number of target sequences are indicated in brackets.

| Target species | *Enterococcus faecalis* | *Enterococcus faecium* | *Pediococcus acidilactici* | *Pediococcus pentosaceus* |
|---|---|---|---|---|
| **Running time** (h:min) | | | | |
| RUCS | 5:17 | 5:20 | 2:11 | 3:16 |
| fdp | 8:32 | 17:44 | 0:36 | 0:02 |
| SpeciesPrimer | 5:23 | 6:53 | 1:35 | 1:11 |
| **Primer pairs** | | | | |
| RUCS | 199 | 107 | 123 | 629 |
| fdp | 6 | 3 | 3 | 6 |
| SpeciesPrimer | 11 | 3 | 2 | 36 |
| **Primer-BLAST** (perfect / total) | | | | |
| Best primer pair | | | | |
| RUCS | specific (59/59) | specific (153/153) | specific (11/11) | specific (9/9) |
| fdp | specific (58/59) | specific (153/153) | specific (10/11) | specific (9/9) |
| SpeciesPrimer | specific (59/59) | specific (153/153) | specific (11/11) | specific (9/9) |

**Notes.**
fdp, find_differential_primers.

The results of the SpeciesPrimer pipeline for the four target species ranged from two to 160 identified primer pair candidates. Several factors can influence the number of identified primer pairs, such as the quality of the input genome assemblies, assemblies with wrong taxonomic labels and the genetic diversity within the species. A low-quality assembly with missing sequences or contaminations can decrease the number of identified core genes. The initial quality control helps to minimize the risk that such assemblies are included in the pipeline runs. However, also an increased sequence diversity, either due to sequencing errors, assembly errors or real diversity, limits the number and the length of identified conserved sequences. Subsequently this affects the yield of reported primer pairs, since the pipeline selects highly conserved sequences for primer design. The two test cases designed to exemplify the influence of the input genome assemblies on the pipeline results illustrate that the SpeciesPrimer pipeline performs best on closely related (same species) genome assemblies with a good overall quality.

The specificity of the designed primers was evaluated *in silico* by BLAST with a more extensive database (RefSeq Genome) than the one used for the specificity check during primer design. The validation showed that the specificity of the tested amplicons was high and no other species than the target species had an identical sequence. Most target sequences in the database showed a perfect match for the primers in the primer-binding region. For all tested primer pairs, only the expected PCR products for the target species and no amplicons for other sequences were predicted by Primer-BLAST. The results of Primer-BLAST indicate that the reported primer pairs were very specific, even though the species list used for the specificity evaluation during primer design covered only 259 non-target species.

In this work, 21 to 25 target strains for each target species and 120 non-target strains have been tested to assess inclusivity and exclusivity of the qPCR assays, respectively. The *in*

*vitro* validation of primer pairs has shown that the *in silico* validation is not always able to predict non-target PCR products. The fluorescence signals occurring at late quantification cycles (Cq >30) are probably due to PCR products with suboptimal primer binding. Testing the qPCR assays in mixtures and communities could be interesting to assess if these PCR products also accumulate in presence of target DNA. The specificity could be sufficient in mixtures due to competition for the primers and the difference in primer binding and amplification efficiency. For many research applications, qPCR assays with a low signal in negative samples are acceptable, assuming that low-level signals can be distinguished from low concentrations of target species DNA by the melting curve analysis (*Ririe, Rasmussen & Wittwer, 1997*). Furthermore, for many applications, the annealing temperature can be optimized by empirical determination of a suitable annealing temperature and the primer concentration can be adjusted to improve the specificity of the assay (*Bio-Rad Laboratories, 2019*). We did not try to optimize our assays with these measures, because the aim was to design primers for high-throughput qPCR, requiring the exact same PCR conditions. For the tested qPCR conditions, the most specific qPCR assays were Ec_faeca_acuI (*E. faecalis*), Ec_faeci_cysS (*E. faecium*), Pd_acidi_asnS (*P. acidilactici*) and Pd_pento_nagK (*P. pentosaceus*). Further work will be necessary in order to make these qPCR assays fully operational for the quantification of bacteria in a complex system such as food. For instance, suitable qPCR standards should be designed and validated, so that the limit of detection of each assay can be determined (*Forootan et al., 2017*).

Primer-BLAST, fdp and RUCS allow designing primers for different applications, but demand experience and manual manipulations. Primer-BLAST designs primers and performs specificity checks, but requires a target sequence provided by the user. In the case of RUCS, manual manipulation and some experience is needed to prepare the positive and negative reference sets. The same applies to fdp and the results from the comparison indicate that fdp has its strength in the identification of strain-specific primer pairs and for subsets of the positive set as implied in the name. The observed mismatches in the primer-binding region are probably due to the alignment-free approach the pipeline uses. This seems to drastically increase the speed, but it is not taking into account the conservation of the target sequence and therefore mismatches, e.g., due to single nucleotide polymorphisms (SNPs), can be found in the primer binding region. For a large number of input assemblies, e.g., for *E. faecalis* (575), fdp requires distinctively more time to run, which is a known issue caused by the cross-validation prediction step using PrimerSearch (*Pritchard et al., 2012*).

Compared to primer-BLAST and RUCS, the task SpeciesPrimer performs is really specialized, namely to design primers for species-specific sequences. In contrast, SpeciesPrimer requires no previous knowledge about the input genome assemblies and no manual manipulation of sequences has to be performed. The ability of SpeciesPrimer to run on standard computers with good performance instead of specialized high-performance computers should allow primer design for the wider range of scientists. Docker containers simplify the installation procedure and should allow non-bioinformaticians to setup and use the SpeciesPrimer pipeline.

## CONCLUSIONS

In this work, we presented the SpeciesPrimer pipeline, which is a fully automated pipeline from the download of bacterial genomes, the identification of conserved species-specific core genes to primer design and subsequent quality control of primer candidates. Primers for four bacterial species were designed and validated and have shown to perform adequately under the same qPCR conditions.

A standard computer with good performance, good quality genome assemblies, a local copy of the nt BLAST database and a list of non-target bacterial species are the only requirements for primer design with SpeciesPrimer. A complete image of a Linux OS with all dependencies and the pipeline scripts is available from Dockerhub. To simplify primer design for users not familiar with command line tools, a graphic user interface is provided in the latest version of SpeciesPrimer. SpeciesPrimer facilitates efficient primer design for species-specific quantification, paving the way for a fast and accurate quantitative investigation of microbial communities.

## ACKNOWLEDGEMENTS

We would like to thank Marco Meola and Remo Schmidt for critically reading the manuscript and many helpful comments and Daniel Marzohl, Nadine Sidler, Elvira Wagner and Kotchanoot Srikham for their valuable technical help.

### Funding
The authors received no funding for this work.

### Competing Interests
The authors declare there are no competing interests.

### Author Contributions
- Matthias Dreier conceived and designed the experiments, performed the experiments, analyzed the data, prepared figures and/or tables, authored or reviewed drafts of the paper, and approved the final draft.
- Hélène Berthoud, Noam Shani, Daniel Wechsler and Pilar Junier conceived and designed the experiments, authored or reviewed drafts of the paper, and approved the final draft.

### Data Availability
Code is available at GitHub:
https://github.com/biologger/speciesprimer
Docker image is available at DockerHub:
https://hub.docker.com/r/biologger/speciesprimer.
WGS data is available at NCBI BioProject: PRJNA576774.

## Supplemental Information

Supplemental information for this article can be found online at http://dx.doi.org/10.7717/peerj.8544#supplemental-information.

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
