# Peer review of "SpeciesPrimer: a bioinformatics pipeline dedicated to the design of qPCR primers for the quantification of bacterial species"

_PeerJ, doi:10.7717/peerj.8544_

## Round 0.1 · original submission · Major Revisions

In your revision, please ensure that the genomic data for the strains in the Agroscope Culture Collection are made available.

Reviewer 1 ·

Basic reporting

"SpeciesPrimer: A bioinformatics pipeline dedicated to the design of qPCR primers for the quantification of bacterial species" by Matthias Dreier et al present a software tool SpeciesPrimer that aims to be a one-stop shop for designing differential qPCR primers. Overall, the manuscript clear, concise, and well-presented. The introduction sets up the need for better tools to design diagnostic primers in a high-throughput manner. In particular, the specific knowledge gap appears to be the need to primer design to be both species-specific, user-friendly, but also constrained to the parameters of microfluidic qPCR systems (ie not requiring optimization steps for cycling conditions, etc). All the code is available on GitHub, and the supplementary data is well organized. Overall, it appears that the pipeline has been put together in a logical manner, and the case study of developing primers for 4 cheese-relevant strains seems strong. The documentation on the GitHub repository seems sufficient to get new users comfortable with the interface (well done with adding the notes to help those new to Docker!). The structure mostly conforms to PeerJ standards, though there is some confusion arising from having both a supplementary Data S1 and Dataset S1.

The figures need a bit of work to more clearly convey the information:
Figure 1: Either the leaf names need to be simplified, the figure enlarged, or both. Vector graphics are preferred (https://peerj.com/about/author-instructions/#figures), and would allow viewers to highlight text, zoom without losing quality, etc. Perhaps the low quality of the images is an artifact of the reviewing system, but regardless, these trees need to be improved for clarity (see guidelines here http://currents.plos.org/treeoflife/index.html%3Fp=6345.html). Including the newick trees in the supplementary data would be nice as well.
Figure 2: This also should be a vector graphic (otherwise I can't search for the strain I am interested in). The color key should be moved next to the color bars on the left-hand side of the figure (or the strain IDs to the left and the color bars to the right), as they do not line up with the data. Lastly, the strains should be labeled as they are in the qPCR data (perhaps in brackets?) -- it is tedious to have to relate the data from Dataset S1 to Data S2 before viewing Figure 5.

The genomic data for the strains in the Agroscope Culture Collection does not appear to be made available. If this is not the case, links must be provided for accessing the data. As it stands, this severely limits the reproducibility of the analysis. In particular, in the case study for P. acidilactici, of the 127 genomes considered, only 9 are available; for P. pentosaceus, only half are available. Given the plethora of other possible genomes to consider, I believe the analysis should only contain genomes publicly available, or the genomes for the FAM* strains be deposited in the public repositories.

That said, I appreciate that the authors included accession number for all strains used, and the raw qPCR data.

Experimental design

The Code:
It would be a good idea to either (a) add citations in a README file in https://github.com/biologger/speciesprimer/tree/master/pipeline/ext-scripts, or to (b) use git submodules to use/credit code not written by the authors. For instance, the DBGeneratory.py script is credited in the manuscript, but not in the repository. Similarly, it would be good to include a README file in the https://github.com/biologger/speciesprimer/tree/master/pipeline/dictionaries directory, explaining how the files there are generated.

- There are no tests for the code, apart from whether a build passes on Dockerhub. I suppose its not required for publication, although in the long run the authors might consider employing a test framework to make maintaining it easier.

The Manuscript:
The pipeline presented is well described. Some readers may wish to see an explanation for tool choice -- for instance, why Prokka for annotation, why Roary for pan-genome analysis, why PRANK for alignment, etc. SpeciesPrimer pipeline primarily leverages published tools, perhaps a table containing the features of the different primer design tools would be useful in the introduction as well.

This is primarily a software paper, but the paper lacks the necessary comparison to existing state-of-the-art tools. There is no shortage of various primer design tools (the "primer design" topic on GitHub has 20 repositories https://github.com/topics/primer-design), and no comparisons are given. A failed comparison with RUCS is mentioned in passing on line 83, but I would like to see what data and parameters were used for this test. Specifically, on line 88 the manuscript says the" RUCS works best for very similar genome assemblies in the positive and the negative sets.... and was therefore not suitable for our high-throughput approach", while later in the manuscript on line 313, "The results of the two test cases illustrate that the SpeciesPrimer pipeline performs best on closely related genome assemblies with a good overall quality"; more information should be provided as to how the two situations differ.

Particular tools that I would like to see a comparison to include:
- https://github.com/quwubin/MFEprimer
- https://github.com/widdowquinn/find_differential_primers
- https://github.com/billzt/PrimerServer

Validity of the findings

As noted above, including non-public genomic data in the analysis prevents reproducibility. Either the data should be made public, or these strains should be excluded from the analysis.

Specific comments:
- 224: Although in the majority of cases your error is pretty small, the use of technical duplicates is concerning. For instance, for the acuL set "nontarget4", you have one Lactobacillus zeae DSM 20178 (FAM 2136) replicate with a Ct value of 12.30, and the other is empty. It appears that these instances are dropped? There are several other examples of single replicates having Ct values that don't appear to be reported. A similar case seems to happen with Lactobacillus buchneri LMG 11439 FAM 23117 for the purD assay. Perhaps consider https://www.ncbi.nlm.nih.gov/pmc/articles/PMC4779984/ for recommendations on choosing a number of replicates (along with their budget walkthrough document https://jdieramon.github.io/BlueberryProject/optimal_Plan.html).
- Further, at first reading I thought "four qPCR runs" meant that these were run with 4 biological replicates, however the raw data appears to show only a single biological replicate, and that the 4 runs means it took 4 plates to run all of the non-target samples. This should be clarified.
- In Supplementary data S1, or in the text, no explanation is given for changing the default pipeline parameters for certain runs. How was this decided?
- 292: it is unclear why the authors calculated ANI for the subclusters of E. faecium, P. acidilactici separately from the whole species. Displaying this as a heatmap might be more helpful to show the clusters. Perhaps they were all calculated together -- if so, Table 3 should be changed to reflect that.
- If the stock for FAM20347 is contaminated, it should be removed the the analysis entirely.
- 331: In Supplementary Data S3, it is unclear how you are defining "anomalous alignments". Why were these sequences excluded?
- 113: I would appreciate a note about database usage. The nt database excludes WGS records (ftp://ftp.ncbi.nlm.nih.gov/blast/documents/blastdb.html), and this may impact the pipeline usage. Noting this caveat and explaining why this does or does not influence the results would be helpful.
- 170: It would be helpful to further explain the target template database, the non-target sequence database, and the target genome databases. Perhaps having a schematic showing the workflow?
- 141: it is unclear why you are citing https://github.com/EnzoAndree/tutorials/blob/patch-1/DBGenerator.py. This is the fork of the repository https://github.com/microgenomics/tutorials, and it appears that they should be given proper credit.
- 96: The paper (and the GitHub repo) list the hardware requirements to be a machine with 16GB RAM and multiple cores, suggesting that it can be run on consumer hardware. The timing benchmarks should reflect that -- the analysis was run on a machine with double the ram and ~5x the CPUs available on an average consumer computer (line 202). Especially as pangenome analysis can be RAM intensive, I would be interested to see if the analysis could be run on a 16GB machine.
- Supplementary table S1: it is unclear why cell E2 is highlighted yellow.
- As SpeciesPrimer includes a tool for updating nt, it is a bit surprising that the last time the db was updated was 20 April 2018.
- Table 1: please note the versions of software used in constructing the Docker image
- Table 2: please note the hardware used for timing
- 219: It might be worth noting why you use the "nr" database here (as there is a discrepancy between normal blast calling "nr" the protein database and primer-blast calling "nr" the non-redundant nucleotide database).

Additional comments

The paper (and software it describes) is well presented, well thought-out, and I believe is close to where it needs to be for publication. My final note is about the usability of the tool

It appears that there is way to access the functionality of the tool from the command line. The browser-based interface is nice, but is cumbersome working on remote systems. For instance, without root access, I can't set up xforwarding on the HPC I work on. So, without a command-line interface, I can't run SpeciesPrimer on the HPC; I don't have space for downloading ~65GB nt database on my personal machine, so trying out the software is currently impossible. One possible solution would be to incorporate a command-line utility program to supplement the GUI for users like myself. It would be a relatively small change that would be both easier to run on shared hardware with HPCs where users don't have root access and also make the execution of the program more reproducible. Another option would be allowing the user to use a custom BLAST database instead of the full nt. Users on consumer hardware could, for instance, use a smaller database.

SpeciesPrimer promises to be a "batteries-included" solution to the primer design problem, and I hope the authors are able to make the recommended changes to remove some of the final barriers to usage.

Reviewer 2 ·

Basic reporting

Overall, well written manuscript with clear English and no grammatical/syntax errors.

According to the author guuidelines, if you are using Mendeley (which I assume you are using), you should change your field codes into plain text by performing the following:
If using Windows:
Ctrl-A (Selects all the text in the manuscript).
Ctrl-Shift-F9 (Changes the fields codes).
If using OSX:
Command-A (Selects all the text in the manuscript).
Command-6 (Changes the fields codes).

Experimental design

The manuscript falls under the Aims and Scope of the journal.
The pipeline is very useful although it would be difficult to apply it to other kinds of samples apart from food (e.g. environmental samples).

Validity of the findings

No comment.

Additional comments

Line 24: Since you are focusing on food microbiology, I would suggest you rephrase the "given system"
and make it more food specific/relevant. It is good that you give the example of cheese, but I would like
to see it a bit more clear before that.

Line 63: You could add information on what these genes encode

Lines 65-66: You can elaborate a bit more.

Lines 67-69: You can also add PrimerMiner (doi: 10.1111/2041-210X.12687)

Line 101: There is no Title (and legend) for your Table. Are you sure you should not include it in the doc file also?

Line 104: How is the specificity for the target species assessed?
If I understand correctly, you donwload genome assemblies for specific species.
A true specificity check would require comparison with genome assemblies for other species also.

Line 166: You should specify where the script is located in your github repository.
Users may not be familiar enough to find it on their own.

Line 142: You should refer better to the script. Don't just put it in the brackets.

Line 147: Rephrase to "using consambig function"

Line 148: Correct to "sequences"

Line 152: Correct to "consensus"

Line 166: Correct "in silico" using italics. Also, in line 214 and line 325.

Line 211: Data S1: The first cell in the "Test case 1 assembly accessions" column is highlighted yellow

Lines 218-220: Why did you choose "BLAST database (nr) limited to sequences from bacteria"
and not another kind of database, e.g. Silva?

Lines 272-273: You can delete it, the reader already knows about it .

Lines 283-284: "were created using Roary and FastTree". You can delete it. This information is already mentioned
in the Materials & Methods section. You should also mention SeaView software in the Materials & Methods section
and not only in the legend of the Figure.

Lines 284-288: Can you re-create the figure and somehow find a way to reduce the node names?
We don't need to see all the node names one by one. They create a lot of noise.

Lines 291-292: This belongs to the Materials & Methods section.

Lines 313-315: This belongs to the Discussion or Conclusions section.

Lines 326: The abbreviation PPC should be explained in the legend Table 4.

Line 331: Maybe you should refer in the specific pages of Data S3 also, to make it easier for the reader.

Lines 393-395: This is very important from my point of view. Did you consider assessing it in your in vitro experiments?

There are no references in the Discussion...
In lines 397-399, for example, you should be able to find a suitable reference.

Lines 401-402: Are you sure this is the appropriate way to cite a website?

---

## Round 0.2 · Major Revisions

The manuscript has been much improved following the first review round and while the improvements are evident and the merit in the work and the software produced recognized, reviewer one has highlighted that the software itself is not currently readily usable. To ensure that the SpeciesPrimer would have the widest reach possible and before consideration for publication I suggest the authors consider reviewer 1's comments to ensure that software can be easily used and common errors addressed.

Reviewer 1 ·

Basic reporting

The authors work to clarify areas flagged in review, restructuring (and expansion) of the supplementary data, updating figures, and other changes have greatly improved the readability of an already good work.

I also appreciate the authors efforts to make the Agroscope genomes available to the reviews and to the community through uploading them to NCBI.

Experimental design

Expanding the manuscript to include a primer design comparison with similar software strengthens the work. I particularly appreciate the care the authors took in document the commands used to allow replication of that analysis.

I am not sure what is meant by the statement “The positive genome sets for RUCS and fdp were the same genome assemblies used for the SpeciesPrimer pipeline. In addition, genome assemblies for 224 different species from the species list were downloaded.” Were these different from those used in the SpeciesPrimer analysis?

Validity of the findings

I appreciate that the authors clearly outlined usage cases for the command line usage with Docker. That said, the docker implementation could be improved. As far as I can tell, it requires the user to enter into the image to actually use it. For instance, attempting to run with docker run results in an error because there is no command line option for email:

```
docker run --rm biologger/speciesprimer speciesprimer.py --target Staphylococcus_aureus --path ./tmp_staph/ --qc_gene rRNA --assemblylevel complete

Start searching primer for Staphylococcus_aureus
To make use of NCBI's E-utilities, Please enter your email address.
fatal error while working on Staphylococcus_aureus check logfile
EOF when reading a line
Error report:
for target Staphylococcus_aureus
Error 1:
fatal error while working on Staphylococcus_aureus check logfile
```
Editing “tmp_db_path” requires interactively editing a config within docker – there is no –configfile argument either, where this could be provided. Further, providing a –configfile arg would allow users to tweak the thresholds of the various tools – especially for pangenome-calling, these thresholds are sensitive and never one-size-fits-all.

That said, I have tried to use the software a few times, and I am of the opinion that more effort must be taken to ensure that it is in a usable condition. Especially as it is geared towards casual users, proper error handling will help the tool get adopted. It looks from the GitHub logs like development is reasonably active, so hopefully some of these issues are being resolved already. I have a few concerns that I will outline based on my (limited) attempts at usage.

When re-running analyses, QC checking is ignored. It appears that the program just checks for the output of the QC results, and doesn’t check whether the run should pass (the default file has headers, so re-running detects a non-empty file and assumes it passed):

```
Preparing files for BLAST
Start BLAST
Run blastn -task megablast -num_threads 4 -query rRNA.part-0 -max_target_seqs 5 -max_hsps 1 -out rRNA_0_results.xml -outfmt 5 -db nt
Error: No genomes survived QC
Run: remove_qc_failures(rRNA)
Error report:
for target Chlamydia_pneumoniae
Error 1:
Error: No genomes survived QC
root@b1f14c5644bf:/home/primerdesign# speciesprimer.py
Create new config files or start pipeline with previously generated files?
type (n)ew or (s)tart:
s
Search for config files for (a)ll or (s)elect targets:
a
Please specify a path to use as the working directory or hit return to use the current working directory:

Search in /home/primerdesign
found: /home/primerdesign/Chlamydia_pneumoniae/config/config.json
Start searching primer for Chlamydia_pneumoniae
12 genome assemblies are available for download

Already annotated:
['GCF_001007085v1', 'GCF_001007105v1', 'GCF_000008745v1', 'GCF_000007205v1', 'GCF_001007045v1', 'GCF_001007065v1', 'GCF_001007145v1', 'GCF_000011165v1', 'GCF_001007025v1', 'GCF_000024145v1', 'GCF_001007125v1', 'GCF_000091085v1']
Found rRNA_QC_report.csv, skip QC rRNA
Start pan-genome analysis
...
```

This run (and 2 others I tried) ended with the following fatal error:

```
22 Oct 2019 12:28:05: Run: run_blast - Start BLAST
22 Oct 2019 12:28:05: Run blastn -task dc-megablast -num_threads 4 -query conserved.part-0 -max_target_seqs 2000 -evalue 500 -out conserv
ed_0_results.xml -outfmt 5 -db nt
22 Oct 2019 12:35:48: Run blastn -task dc-megablast -num_threads 4 -query conserved.part-1 -max_target_seqs 2000 -evalue 500 -out conserv
ed_1_results.xml -outfmt 5 -db nt
22 Oct 2019 12:43:20: > Blast duration: 0:15:15
22 Oct 2019 12:43:21: Run: run_blastparser(Chlamydia_pneumoniae), conserved sequences
22 Oct 2019 12:43:21: Run: blast_parser
22 Oct 2019 12:43:21: Run: blastresults_files(Chlamydia_pneumoniae)
22 Oct 2019 12:43:29: fatal error while working on Chlamydia_pneumoniae check logfile
fatal error while working on Chlamydia_pneumoniae
Traceback (most recent call last):
File "/home/pipeline/bin/speciesprimer.py", line 4261, in main
config).run_blastparser(conserved_seq_dict)
File "/home/pipeline/bin/speciesprimer.py", line 2611, in run_blastparser
align_dict = self.blast_parser(self.blast_dir)
File "/home/pipeline/bin/speciesprimer.py", line 2496, in blast_parser
blastrecords = self.parse_BLASTfile(filename)
File "/home/pipeline/bin/speciesprimer.py", line 2184, in parse_BLASTfile
record_list = list(blast_records)
File "/usr/local/lib/python3.5/dist-packages/Bio/Blast/NCBIXML.py", line 626, in parse
expat_parser.Parse(NULL, True) # End of XML record
xml.parsers.expat.ExpatError: no element found: line 603955, column 0
22 Oct 2019 12:43:29: > Error report:
22 Oct 2019 12:43:29: > for target Chlamydia_pneumoniae
22 Oct 2019 12:43:29: > Error 1:
22 Oct 2019 12:43:29: > fatal error while working on Chlamydia_pneumoniae check logfile
```

The log file provided no additional information about what caused the element not found error when parsing the blast results. I have attached the log file to the review (peerj only allows pdf files, so I had to convert it, sorry!), and here is the config, for reference:
```
cat ./Chlamydia__pneumoniae/config/config.json
{"target": "Chlamydia_pneumoniae", "offline": false, "path": "/home/primerdesign", "mfethreshold": 90, "remoteblast": false, "nolist": false, "probe": false, "exception": "Serratia_marcescens", "assemblylevel": ["complete"], "blastseqs": 1000, "qc_gene": ["rRNA"], "skip_download": false, "mfold": -3.0, "ignore_qc": false, "maxsize": 200, "skip_tree": false, "mpprimer": -3.5, "intermediate": false, "minsize": 70, "blastdbv5": false}
```

As I cannot get the software to run (whether due to user-error and no error-catching, or due to an actual fault in the code), I cannot recommend the work to be published. There is a good chance that I have something configured incorrectly, but no error messages help me fix that, then I cannot use the tool.

My concerns regarding qPCR with only single biological replicates and duplicate technical replicates remain. I realize that this is considerable amount of work to redo, but I would rather fewer strains being screened with sufficient power than an underpowered cursory screen of many strains. While I largely agree with the 4 points presented by the authors in the response, the very fact that there was variability that did not show conclusive results highlights the benefit of having both biological and technical replication. Yes, biological replication in a perfect scenario would not contribute new information, however this is common practice to control for contamination of the culture or other handling errors – this is valuable information. I would recommend the editor allows the authors sufficient time to run the necessary experiments. That said, I recognize the author’s concern that this is indeed primarily a software paper, so I would defer to the editor if they feel that the data presented sufficiently supports the purpose of the work.

Additional comments

The authors have made strides towards making SpeciesPrimer a useful tool for the community, and I hope that the changes I suggest here aren’t too arduous. SpeciesPrimer will be a good resource for the community. I would suggest asking a few colleagues to try it out, to help identify common user errors.

Annotated reviews are not available for download in order to protect the identity of reviewers who chose to remain anonymous.

Reviewer 2 ·

Basic reporting

No comment.

Experimental design

No comment.

Validity of the findings

No comment.

Additional comments

Delete the phrase "The phylogeny was created with Roary and FastTree 2 and displayed with iTOL
(https://itol.embl.de/)" from the legends of Figures 2 and 3. The reader already knows this information
from lines 285-286 of the manuscript.

Figure 4: Recreate the figure using other colors instead of green and red.
You should choose color blind friendly colors.

---

## Round 0.3 · accepted · Accept

Thank you for taking the time to address the comments of reviewer one and investigate the source of errors with the software. I'm sure you agree that ensuring it works smoothly prior to publication will increase the usability and adoption of your work within the field.